# Digital Twins in Built Environments: An Investigation of the Characteristics, Applications, and Challenges

**Muhammad Shahzad** [1], **Muhammad Tariq Shafiq** [2,*] , **Dean Douglas** [3,4] **and Mohamad Kassem** [3]

1   Foundation Property Management (FPM), Abu Dhabi 126735, United Arab Emirates;
    shahzad_224485@yahoo.com
2   Department of Architectural Engineering, College of Engineering, United Arab Emirates University,
    Abu Dhabi 15551, United Arab Emirates
3   Department of Mechanical and Construction Engineering, Northumbria University,
    Newcastle Upon Tyne NE1 8ST, UK; dean.douglas@bimacademy.global (D.D.);
    mohamad.kassem@northumbria.ac.uk (M.K.)
4   BIM Academy, Newcastle Upon Tyne NE1 3NN, UK
*   Correspondence: Muhammad.tariq@uaeu.ac.ae

**Abstract:** The concept of digital twins is proposed as a new technology-led advancement to support the processes of the design, construction, and operation of built assets. Commonalities between the emerging definitions of digital twins describe them as digital or cyber environments that are bidirectionally-linked to their physical or real-life replica to enable simulation and data-centric decision making. Studies have started to investigate their role in the digitalization of asset delivery, including the management of built assets at different levels within the building and infrastructure sectors. However, questions persist regarding their actual applications and implementation challenges, including their integration with other digital technologies (i.e., building information modeling, virtual and augmented reality, Internet of Things, artificial intelligence, and cloud computing). Within the built environment context, this study seeks to analyze the definitions and characteristics of a digital twin, its interactions with other digital technologies used in built asset delivery and operation, and its applications and challenges. To achieve this aim, the research utilizes a thorough literature review and semi-structured interviews with ten industry experts. The literature review explores the merits and the relevance of digital twins relative to existing digital technologies and highlights potential applications and challenges for their implementation. The data from the semi-structured interviews are classified into five themes: definitions and enablers of digital twins, applications and benefits, implementation challenges, existing practical applications, and future development. The findings provide a point of departure for future research aimed at clarifying the relationship between digital twins and other digital technologies and their key implementation challenges.

**Keywords:** BIM; digital twin; Internet of Things; construction; smart cities; data security; standardization; cultural change; pilot projects

## 1. Introduction

The industry-wide acceptance of building information modeling (BIM) has accelerated the efforts to digitalize existing processes within the construction sector. BIM has emerged as a technology-led process, offering numerous opportunities to improve decision making during the design, construction, and management of built assets [1,2]. More recently, industry and research efforts have been exploring new technological advancements, such as big data, cloud computing, artificial intelligence (AI), virtual reality (VR), augmented reality (AR), Industry 4.0, and Internet of Things (IoT), to further extend BIM capabilities with enhanced data-driven applications [3]. For example, asset management is one area where such attention has been focused with the aim of transforming existing document-oriented asset management practices into model-based, data-driven, automated processes

of asset management [4]. Numerous research efforts and standards have also addressed the use of BIM for enhanced facility management (FM) and asset management [5,6]. The rationale driving these studies is based on the criticism that BIM applications are far more developed in design compared to their applications in construction and operation due to challenges linked to interoperability [7–10].

The concept of a digital twin is not new. It is well-known in digital simulations across industries such as manufacturing, aerospace, and automobile industries; however, its application to the built environment is a recent development [11]. Built environments refer to one integrated and holistic concept of human-made surroundings that include the planning, design, management, maintenance, and monitoring of functional and physical characteristics in related buildings and infrastructure assets. A recent interpretation of the digital twin concept within the built environment research relates it to the use of digital models of an asset to provide simulations and to an information link to a real-world entity (i.e., a physical twin), thus enabling opportunities for data-centric decision making in asset operations and management [12]. Recent studies suggest the applications of digital twins may play a crucial role in the digitalization of asset creation, delivery, and management in built-environment projects. For instance, David et al. [13] and Grieves [14] claimed that a digital twin facilitates real-time synchronization between a real-world model (physical model) and its virtual copy for improved energy monitoring, prediction, and efficiency enhancement; thus, it can significantly reduce the overall energy consumption.

Some scholars extend the connotation of a digital twin in terms of its coverage of the physical continuum. In this vein, Grieves [14] defines the digital twin as a set of virtual information constructs that fully describes a potential or an actual physically manufactured product from the micro-level (atomic level) to the macro-level (geometrical level). Ideally, any information obtained by inspecting a physically manufactured product can be obtained from its digital twin. Another expansive definition, although its connotation extends the concept from a different facet, is the one proposed by Haag and Anderl [15] who define a digital twin as a complete digital representation of a product or an asset that represents the properties, condition, and behavior of the real-life asset through digital models and data. However, this definition overlaps with the definition of a building information model, which was defined by several authors as a representation of the complete functional and physical characteristics of an asset in an object-oriented data repository [16,17]. This casts doubt on whether a digital twin is different from a building information model, which is also an issue of contention in both industry and academia. This has led to the following research question: What are the common characteristics and definitions of a digital twin for the construction industry and what are seen as its potential applications in construction project delivery and built asset management? Moreover, it is unclear how digital twins relate to existing digital technologies (e.g., BIM, VR, AR, IoT, AI, and cloud computing) used in construction projects.

With this background, the objective of this study is to examine the concept of digital twins and investigate their potential applications and challenges within built-environment projects. The remainder of this paper is organized as follows: Section 2 describes the research methodology and provides details of the semi-structured interviews conducted to collect and analyze the research data; Section 3 presents a review of the literature exploring the concept of digital twins, presenting arguments on the definition and characteristics of digital twins, and highlighting potential applications and challenges from previous works; Section 4 presents research results and discusses the results based on five themes extracted from the analysis of the semi-structured interviews and literature review; and Section 5 concludes this paper and sets the directions for future work.

## 2. Research Methodology

Digital twins are relatively new in the architecture, engineering, construction, and operation (AECO) industry, and therefore several ambiguities exist in terms of the concept, applications, and integration of digital twins with existing technologies used in the industry.

In such situations, where the problem is both new and not well researched, an exploratory research design is appropriate. Hence, this paper adopts an exploratory research methodology and employs a combination of literature review and semi-structured interviews. In the literature review (Section 2), relevant studies on digital twins were examined. Questions for the semi-structured interviews were derived based on the literature review.

A semi-structured interview is a suitable data-collection methodology for exploring the opinions of research participants on information and problems and eliciting descriptive responses to research questions. This method was applied in this study to seek expert opinion on the following questions: (1) What is a digital twin for built environments? (2) What is the relationship of digital twins with BIM and other associated technologies? (3) What are the potential applications and challenges for the implementation of digital twins? (4) What can foster the development and implementation of digital twins in the built-environment sector? The semi-structured interviews were conducted following the method described by Redmond et al. [18]. The research objectives were transformed into five themes, which were identified from the literature, and relevant questions were added for each theme. The themes are as follows:

1.  Theme 1: Definitions of digital twins, enablers, and relevance considering the existing technologies. This theme included questions about the opinions of the participants on digital twins and their potential relationship or overlap with BIM.
2.  Theme 2: Applications and benefits of digital twins. This theme included questions aimed to explore the potential of digital twins and solicit the opinion of the participants about the desirable characteristics and requirements of digital twins within the built environment context.
3.  Theme 3: Digital twin implementation challenges. This included questions related to the various challenges in the development and implementation of digital twins. As the selected interview participants represented different industry roles, the questions in this category were intended to obtain different perspectives for describing the perceived challenges for applications of digital twins in the AECO industry.
4.  Theme 4: Existing practical experiences of digital twins. All the research participants were highly experienced professionals in the AECO industry and were involved in the implementation of BIM in different capacities. Therefore, this theme aimed to capture some evidence about or the status of practical applications of digital twins through the interviewed expert, if they had any experience using digital twins.
5.  Theme 5: Future improvement suggestions and timeline. This category included questions seeking to understand different opinions on the future of digital twins and the steps that can be taken to facilitate the development and implementation of digital twins in the AECO industry.

In this study, a purposive sampling technique was employed. This technique is an effective sampling technique when investigating a problem that requires in-depth and detailed information about the phenomenon under investigation. It is suitable for the investigation of multi-faceted topics such as the adoption of digitalization technologies in the built environment [19,20]. A list of industry experts was prepared using personal connections and by scanning through the LinkedIn profiles of related industry experts. Potential interview candidates were identified by reviewing their industry experience, sector of employment, educational background, and experience with digitalization in the AECO industry. Interview invitations were sent to selected candidates who were representatives from various roles in the AECO industry. Finally, interviews were scheduled and conducted using online digital communication platforms (Skype and Zoom), as face-to-face interviews were not possible owing to the standard operating procedures imposed by the COVID-19 pandemic regulations. A summary of interview participants is presented in Table 1.

**Table 1.** Details of the participants.

| No | Description | Current Position | Experience (Years) | Qualification |
|---|---|---|---|---|
| 1 | Participant 1 | Vice Chair of International Consultancy | 40 | Architect/IT expert |
| 2 | Participant 2 | BIM Manager (Consultant) | 14 | Civil engineer with M.Sc. in Engineering |
| 3 | Participant 3 | Senior Manager, Virtual Design and Construction in a contracting organization | 15 | Master's in Building and Construction Management |
| 4 | Participant 4 | Senior Digital Delivery Specialist (Government Construction Regulation Agency) | 11 | M.Sc. Mechanical Engineering |
| 5 | Participant 5 | Head of Automated Design and BIM | 16 | B.Eng. in Civil Engineering |
| 6 | Participant 6 | Director, Public Sector Affairs | 28 | M.Sc. in Facilities Management |
| 7 | Participant 7 | Senior Programme Manager | 25 | M.Sc. in Facilities Management |
| 8 | Participant 8 | Principal Consultant | 13 | B.S. in Geography |
| 9 | Participant 9 | Global Construction Practice Director | 16 | M.Sc. in Project Management |
| 10 | Participant 10 | Engineering Information Manager | 32 | |

The semi-structured interviews were recorded and transcribed. The interview transcripts were analyzed using a thematic analysis approach, which allowed the systematic analysis of interview text data using qualitative data analysis software (NVivo 12 Pro). Using the software, the collected data were thoroughly studied and segregated using a coding structure. The production of codes led to the generation of themes by combining different codes based on the similarity of their content. Thus, the five categories/themes, which were described earlier in this section, were obtained. Each theme was examined based on the data derived from the deliberations of the study. The commonalities and differences between different participants against a particular theme were obtained and analyzed. Finally, the themes were discussed, reflecting upon the similarities and contradictions between the participants' views and literature review.

## 3. Literature Review

The concept of a digital twin is well established in several industries (e.g., the manufacturing, aerospace, and automobile industries). However, it is relatively new in the AECO industry. Many definitions of the digital twin for the built environment and AECO industry have been proposed, but there is no consensus regarding the definition. For example, the Centre for Digital Built Britain defines a digital twin as a realistic digital representation of a physical entity [21]. Batty [22] defines the concept as a mirror image of a physical process expressed alongside the process and usually matching the physical operation exactly and in real-time. Brilakis et al. [12] define a digital twin as the digital replica of a physical built asset.

A comparison of different definitions of the term "digital twin" is presented in Table 2.

**Table 2.** Various definitions of the term "digital twin" in literature.

| No. | Definition | Authors |
|---|---|---|
| 1 | A digital twin is a realistic digital representation of assets, processes or systems in the built or natural environment. What distinguishes a digital twin from any other digital model is its connection to the physical twin. Based on data from the physical asset or system, a digital twin unlocks value principally by supporting improved decision making which creates the opportunity for positive feedback into the physical twin. | [21] |
| 2 | A digital twin is a digital replica of a physical built asset. What a digital twin should contain and how it represents the physical asset are determined by its purpose. It should be updated regularly to represent the current condition of the physical asset. A digital twin should be standardized yet extensible, able to address key use cases directly and specialty use cases with extensions, cloud and computationally friendly, scalable, and verifiable. | [12] |
| 3 | A digital twin is a mirror image of a physical process that is articulated alongside the process in question, usually matching exactly the operation of the physical process which takes place in real time. | [22] |
| 3 | The digital twin is a hierarchical system of mathematical models, software services, and computational methods, which facilitates real-time synchronization between a real-world model (physical model) and its virtual copy for improved monitoring to the efficiency of the equipment. | [13] |
| 4 | A Digital Twin is an integrated multiphysics, multiscale, probabilistic simulation of an as-built vehicle or system that uses the best available physical models, sensor updates, fleet history, etc., to mirror the life of its corresponding flying twin. | [23] |
| 5 | A digital twin is a computerized model of a physical device or system that represents all functional features and links with the working elements. | [24] |
| 6 | The digital twin is actually a living model of the physical asset or system, which continually adapts to operational changes based on the collected online data, information, and can forecast the future of the corresponding physical counterpart. | [25] |
| 7 | A digital twin is a virtual instance of a physical system (twin) that is continually updated with the latter's performance, maintenance, and health status data throughout the physical system's lifecycle. | [26] |
| 8 | Digital twin is a set of virtual information that fully describes a potential or actual physical production from the micro atomic level to the macro geometrical level. | [27] |
| 9 | A digital twin is the combination of a computational model and a real-world system, designed to monitor, control and optimize its functionality. Through data and feedback, both simulated and real, a digital twin can develop capacities for autonomy and to learn from and reason about its environment. | [28] |
| 10 | Digital Twin is the collection of relevant digital artefacts that involves engineering and operation data, in addition to behavior description using various simulation models. | [29] |

Brilakis et al. [12] highlighted that all definitions of a digital twin include a physical model, a digital model, and an information link between the two. Some authors have used the type of data link between the physical and digital parts to draw a distinction between a digital model, a digital shadow, and a digital twin. For example, Kritzinger et al. [30] and Fuller et al. [31] state that a digital model virtually represents the functional or physical

characteristics of a physical model and may not have any automated data exchange with the physical entity. Furthermore, if there exists a one-way data flow from the physical model to the digital model, the digital model is called a digital shadow; if an integrated bidirectional data flow exists, then the digital model is called a digital twin.

Haag and Anderl [15] highlighted that, contrary to common understanding, a digital twin may not be a collection of all the digital artifacts accumulated during the production of a physical twin. These authors cited Boschert and Rosen [29] who stated that a digital twin is essentially a collection of only those relevant data that may serve the intended purpose of digital twin creation. Madni et al. [26] extended this explanation and described five levels of digital twins depending on the level of development and characteristics of the digital twin: a level 1 digital twin is the conventional prototype produced for engineering purposes and supports early decision making at the concept stage. A level 2 digital twin can incorporate historical data, maintenance data, and operational performance from the physical twin. A level 3 digital twin is the adaptive digital twin that contains an adaptive user interface for the digital and physical twins; it can learn the priorities and preferences of the operator in various contexts and is continuously updated based on the information received from the physical part in real time. A level 4 digital twin is an intelligent model that includes the functionalities of level 3 and is capable of self-learning without any supervision. Finally, a level 5 model is receptive to the data provided by external sources and processes the data for analysis in a situational context.

However, what is apparent is that there seems to be an agreement among all understanding that (1) a digital twin is a representation of a physical asset which represents a physical entity and (2) a digital twin must be coupled with the physical part so it can evolve to reflect its changes. For the AECO industry, the physical part refers to built assets (e.g., residential and commercial buildings, hospitals, bridges, tunnels, industrial factories), and the digital part refers to a three-dimensional (3D) model containing information that can be linked to the physical part, although not all definitions and interpretations of digital twin require the digital replica to be in the form of a 3D model. According to Brilakis et al. [12] the type of data link between the digital and physical twins depends on the purpose and functionality of the digital twin. For example, a digital twin of a construction site may be developed to show the progress of work, and thus the related data link will be established to capture the site data and reflect the up-to-date construction progress in the digital twin. Similarly, a digital twin developed for facilities management purposes will require data of different types and forms, and the link with its physical twin will involve a different frequency of data synchronization. Brilakis et al. [12] conclude that a digital twin in the built environment is a 3D model-based representation of a physical asset (i.e., a physical twin), which may be created to serve a specific purpose in a project lifecycle and would contain the related data required to simulate the realistic characteristics of the physical twin within the defined use case of the digital twin. In addition, a digital twin is not necessarily a single model; it could be a combination of models or data instances, located independently but connected logically, representing a federated digital twin of a physical twin.

### 3.1. Building Information Modeling and Digital Twins

Douglas et al. [32] highlighted some conceptual overlap between the concepts of digital twins and BIM and argued that understanding how the two concepts relate to each other is essential for the future development of digital twins within the built environment. They identified three understandings that are prevalent within the literature: digital twin as a continuation and advancement of BIM; BIM and digital twins as two distinctly separate concepts due to a number of succinct differences; and BIM and digital twins as two complementary concepts where one can be utilized to enrich the other. These groupings of understanding are also seen in various studies. For example, Khajavi et al. [9] claimed that a building Information Model is tuned for the design and construction stages and not for operational and maintenance purposes. A digital twin is developed to monitor a physical asset (under construction or constructed) and analyze the functionality of the installed

equipment to improve operational efficiency through the predictive maintenance of the building [3,21]. A building information model can be specifically developed to embed information in a 3D model with additional information related to equipment specifications, cost estimations, time schedules, and operation and maintenance management. This information has the potential to lay the foundation for the development of a digital twin and its utilization for regular monitoring and maintenance activities through a central collaborative network [9,33]. Regardless of the debate on whether a building information model can fulfil the requirements of operations and maintenance, the data held within a BIM model can be greatly beneficial if integrated into a digital twin. In line with this proposition, Khajavi et al. [9] propose a process for developing a digital twin from BIM (Figure 1).

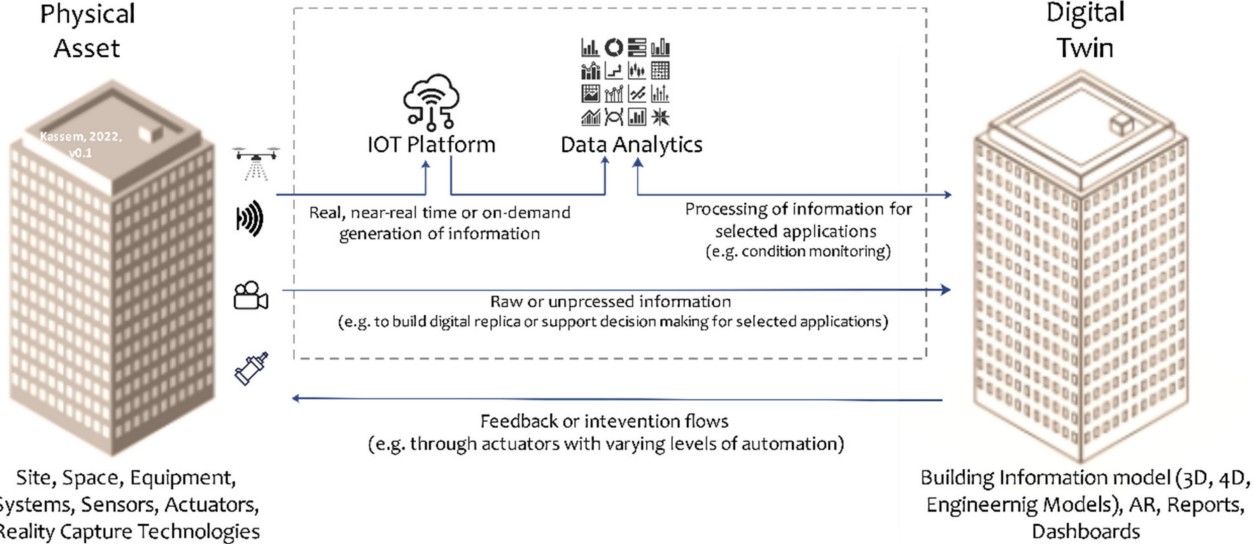

**Figure 1.** Essential components to create a digital twin of a building (adapted from ref. [9]).

A comparison of the characteristics of BIM and digital twins is presented in Table 3 based on information derived from the literature.

**Table 3.** Characteristics of BIM and digital twin as reported by different studies.

| Characteristics | BIM | Digital Twin | Authors |
|---|:---:|:---:|:---:|
| 3D modeling–visualization | ✓ | ✓ | [9,34] |
| Creating a real-time virtual model | ✕ | ✓ | [9] |
| Live model updates through sensors | ✕ | ✓ | [9,34] |
| Data exchangeability between virtual and physical models (two-way communication) | ✕ | ✓ | [35] |
| Data standardization | ✓ | ✓ | [36] |
| Scheduling | ✓ | ✓ | [9] |
| Major contribution at design stage | ✓ | ✓ | [18,26,36] |
| Contribution at construction stage | ✓ | ✓ | [3,36] |
| Major contribution at operations stage | ✕ | ✓ | [26,34,36] |
| Increased collaboration | ✓ | ✓ | [26] |

**Table 3.** *Cont.*

| Characteristics | BIM | Digital Twin | Authors |
|---|:---:|:---:|:---:|
| Time management | ✓ | ✓ | [35] |
| Budget management | ✓ | ✓ | [9] |
| Project simulation analysis | ✓ | ✓ | [9,26] |
| Project simulation analysis in context with surroundings | ✕ | ✓ | [15,34] |
| Live monitoring of assets | ✕ | ✓ | [34] |
| Live and instant updates on equipment status | ✕ | ✓ | [34,35] |
| Instant response to equipment failures | ✕ | ✓ | [28] |
| Realistic predictive maintenance | ✕ | ✓ | [9] |
| Getting insights to improve building utilization and performance | ✕ | ✓ | [34] |
| Reduced project time and cost over project lifecycle | ✓ | ✓ | [37] |
| Easy application on existing buildings | ✕ | ✓ | [9] |
| Better value for employers | ✓ | ✓ | [28] |
| Improved building sustainability | ✓ | ✓ | [36] |
| Improved dynamic risk management at construction site | ✕ | ✓ | [28] |
| Enhanced site logistics | ✕ | ✓ | [38] |
| Updated data for Operation & Maintenance (O&M) purposes | ✕ | ✓ | [26,37] |
| Use of machine learning and automated processes | ✕ | ✓ | [9] |
| Use of self-learning algorithms | ✕ | ✓ | [3,9,26] |
| Necessary use of CDE | ✕ | ✓ | [9,36] |

In summary, there are several differences in the development and abilities of BIM and digital twins that raise several questions around the various definitions, relationships, and integration. Douglas et al. [32] outlined that there is no prevalent understanding or views about the differences or overlap between the two concepts, with all parties having multiple sources of supporting as well as contrary evidence. Despite this lack of a consensus, the development of a digital twin will undoubtedly be influenced by BIM and the journey of its adoption into the AECO industry [32].

### 3.2. Building Management System/Building Automation System and Digital Twin

The functionalities of a digital twin are currently being exercised by various combinations of systems such as the building management system (BMS) and building automation system (BAS) in facility management (FM) practices [37,39]. One of the premises for using digital twins for FM is to foster better collaboration across different disciplines for space management and to digitize operations and maintenance. However, the current BMS and BAS lack integration with the existing 3D modeling practices for FM, including digital twins [11,40]. According to Store-Valen [39] this lack of integration is one of the key reasons property managers rely only on BMS and BAS for asset management. This clearly indicates the inability of current practices to link or merge FM systems with digital twins. The BMS data can be used to create a framework that enables digital twins; however, the integration

of BMS with a digital twin-enabled network requires the technical capability of accepting mobile instructions and providing instantaneous and continuous feedback about constantly changing environmental conditions [41].

Current BIM can both reproduce historical data and incorporate real-time data for the simulation and prediction of future conditions and improvements. However, these models are criticized for their limited ability to self-learn, achieve a high level of autonomy, and process data from external sources [36]. A building information model is principally considered for the design and construction stages, whereas digital twins contribute to boosting the operational efficiency of the assets as they facilitate predictive maintenance through the analysis of real-time situations [3,21]. A digital twin offers more data-driven decision making with its ability to conduct "what-if" analyses during the operation and management of assets compared with a BMS or traditional document-oriented facility management [9].

### 3.3. Applications of Digital Twins

Given the lack of consensus around the differences and commonalities between BIM and digital twins, the next logical step is to review the applications of digital twins reported within the literature. However, this should not be done without highlighting potential areas of contention when crediting an identified application to either digital twins or BIM. This step is performed in this section. As digital twins are a relatively new concept in built environments and in practice, there is still limited literature on their potential applications. The applications identified are described in the following subsections.

### 3.3.1. Smart Cities

Digital twins are reported to have a range of applications and use cases in smart city development with anticipated benefits. For example, the data collected from IoT sensors and embedded into central services within a city can help create state-of-the-art AI algorithms, which can be used for digital twin-enabled city management applications [38,42]. This can result in better traffic management and reduction in congestion and carbon emissions, contributing to the development of sustainable cities [21,31]. A smart city network may include buildings, roads, public services, logistics, people, and power grids, all of which can benefit from the applications of data-driven decision making using digital twins. However, the development of city-level digital twins is at a conceptual stage, and better integration of digital twins with associated digital technologies (IoT, Industry 4.0, Big Data, etc.) [43,44] is required.

### 3.3.2. Design Decision Making

Çıdık et al. [45] and Rasheed et al. [41] argued that current modeling BIM technologies cannot accommodate the dynamism required for design development and improvement; hence, the use of these technologies may be an ineffective method for design processes. Further, Ferguson et al. [8] showed that design development through digital integration has limitations in handling the complexities of the physical world. Hence, designs should be seamlessly tested against multiple parameters, such as energy, thermal, lighting, acoustics, and indoor air quality, to realize greater accuracy and certainty. In this regard, a two-way data flow between the design and related parameters is essential for effective design decision making, thus giving rise to the need for a digital twin application. A digital twin can allow the evaluation of alternative design options at the ideation phase; the design concepts that fail to satisfy the intent of design in accordance with clients' set criteria or other compliance checking can be discarded [35,46]. Compliance checking may include checking for adherence to sustainability requirements or local building laws. Digital twins can facilitate better design decision making by providing the opportunity to test design intent for its functionality or compliance [47]. While it is believed that digital twins have the potential to test and evaluate design iterations against parameters, it is also argued that they cannot be referred to as digital twins as they lack a physical asset to receive data from.

Hence, these claimed applications suffer from a lack of clarity in the distinction between BIM and digital twin capabilities.

### 3.3.3. Product Manufacturing

Applications of digital twins are well known in the product manufacturing sector, as the manufacturing industry has been using digital twins for automatic product manufacturing, driving leaner processes, predictive analytics, and continuous improvement to ensure that as-built enhancements can be implemented in future components [35,48]. The use of digital twins facilitates the production of high-fidelity virtual models; thus, waste-free components can be fabricated, and smart manufacturing processes can be augmented. Furthermore, the use of digital twins will curtail the production of unwanted items and result in sustainable construction.

### 3.3.4. Real-Time Construction Progress Monitoring

Digital twins can play a pivotal role in the digitalization of construction site management by dynamically automating resource allocation and waste management, and by reproducing the real conditions of the site, materials, machinery, and even workers' behavior in a virtual replica [49]. Lavrentyeva et al. [50] stated that continuous tracking and monitoring of materials and human activity is essential, and it requires a fully integrated environment equipped with cameras and sensors with advanced computing power; unfortunately, the construction industry lacks such environments. One of the often-cited differences between BIM and digital twins is the ability of the latter to integrate live data sources. It is argued that BIM processes and technologies cannot incorporate live sensor integration in real practice [51] whereas a digital twin-enabled setup can accommodate such features to ensure the real-time and continuous monitoring of assets. However, this inability to integrate live data feeds into a building information model is also a disputed claim as there are a number of examples that contradict this, such as in Alves et al., [52]; Chen et al., [24]; and Riaz et al., [53].

### 3.3.5. Facility Management

Dixit et al. [7] claimed that the maintenance and operation of built-environment assets are the major application areas for digital twins. A BIM alone cannot automate FM operations because of both interoperability issues and a development approach focused on providing models for design and construction purposes with limited data for FM purposes [4,9]. The type and form of data required for a digital twin developed for FM (such as one based on COBie) will differ from those digital twins developed for other purposes, for instance those developed for use in the design and constructions phases of an asset's lifecycle. Similarly, the data synchronization between the digital and physical twin may differ in attributes such as the update method, level of detail being captured, and the frequency of the of update; these attributes are dependent on the intended purpose of a digital twin [12]. Therefore, a digital twin is intended to offer improved data-driven decision making during asset operation and management.

### *3.4. Challenges*
### 3.4.1. Data Security and Ownership

Cybersecurity is a major concern, especially in web-based environments. Cyber threats such as access to confidential information are a serious risk [26], especially in security-minded projects such as government-owned assets or digital twins at the city level. Data privacy and ownership, which require the definition of access levels and permissions, are also primary and outstanding issues for digital twins [54]. Issues related to intellectual property rights and legal considerations surrounding digital twins should be addressed by designating roles and responsibilities and defining the data accessibility limitations of the participating stakeholders [26]. Although security and ownership issues will have a huge impact on the diffusion of digital twin technology, CDBB [21] outlines that they are

unlikely to greatly hinder the growth of digital twin development. CDBB's National Digital Twin program encourages the creation of digital twins by providing guiding principles and supporting tools for organizations to utilize, update, and adapt on their journey of the development and implementation of digital twins. The CDBB's security principle requires digital twins to be secure by design to enable the protection of personal data and privacy, protection of sensitive national infrastructure assets, protection of commercial interests and intellectual property, and mitigation of risks arising from data aggregation. Their Gemini principles aim to establish a foundation for achieving the ultimate goal of creating an ecosystem of connected digital twins [21].

3.4.2. Lack of Common Data Standards and Tools

Common data standards and interoperability are important enablers for digital twin development. Currently, the development of digital twins is challenged by the lack of consensus on the different standards, technologies, and procedures that can be used to implement digital twins [55]. This issue is also inherent in the enabling technologies of a digital twin. For example, data sharing and the interoperability of digital models are key obstacles in achieving a comprehensive and functional common data environment (CDE) [21,56]. Open standards are fundamental to ensure that digital twin development is vendor-agnostic [57]. The provision of job-specific tools for storing, accessing, and modifying information is fundamental for implementing the digital twin processes. Qi et al. [25] and Lu et al. [3] found that the inability of existing tools to simultaneously integrate as required by the digital twin application is mostly attributed to varying standards, formats, and protocols. The absence of common standards was also found by Re Cecconi et al. [37] as a barrier to the effective implementation of digital twins in FM. Hence, there seems to be a consensus about the need for developing common working standards and tools to facilitate the development and implementation of digital twins in the built environment.

3.4.3. Diversity in Source Systems

The integration of various models with different parametric values, spatial values, and time scales into the digital twin remains a challenge [58]. This hinders the ability to present virtual models that provide a realistic and objective description of the physical assets [34] Qi et al. [48] pointed out that traditional databases are not able to cope with the increasing heterogeneity and volume of digital twin data received from multiple sources. In addition, the problem of reconciling the differences in the semantics and syntax of data is another challenge [59]. Therefore, reaching a consensus on the use of similar tools and a comprehensive database system for efficiently exchanging and managing information are crucial [59]. Interestingly, the challenges to digital twin development and implementation, as revealed by the literature, appear to be similar to those involved in the adoption of BIM practices in the AECO industry. In future, these challenges should be explored in more detail with a more structured classification, such as by grouping them into technology-, process-, policy-, and people-related challenges [60–62].

## 4. Research Findings and Discussion

### 4.1. Theme 1: Definitions and Technology Enablers of Digital Twins

The research results revealed certain common characteristics among the participants in relation to the digital twin concept. The participants opined that a digital twin is essentially a virtual replica of a physical asset capable of understanding and mimicking the operation and usage of an asset. Participants also argued that a digital twin can include, in addition to the data (e.g., maintenance, output, performance) about the concerned asset, some contextual data from the wider environment surrounding the asset. Participants also emphasized the significance of the two-way interaction that occurs between physical and virtual assets which should regularly communicate and share information for up-to-date decision making.

One participant added that digital twins in the built environment are all about using sensors, stating that "sensors are essential for enabling live data feed between the asset models." The participants emphasized that the use of digital twins in the built environment depends on the use of effective tools for information communication, such as a CDE, IoT-enabled devices, and sensor-based data-capturing devices, especially during the asset monitoring and management phases. Two participants stressed that BIM models are intended for design purposes and hence contain limited data; this creates opportunities for the development and use of digital twins in built-environment projects. However, this view was contradicted by three participants, who argued that a building information model and digital twin are essentially identical, as both comprise a 3D model populated with the asset metadata and may be called by any name depending on its potential applications. Four participants also highlighted that the integration of a building information model with computer-aided facility management (CAFM) or a building management system (BMS) can provide the stated functionality of a digital twin, but this would likely be termed BIM for FM and not be labelled with a new term (i.e., a digital twin). Two of the participants noted that CAFM/BMS systems currently lack the ability to integrate or make use of BIM. It was also said that this is often due to the type of information being held within the BIM, which does not pertain to or maintain its relevance to the day-to-day operation of an asset. One further opinion was that "a digital twin is just a better BMS" that can be, for example, programmed to maintain temperatures in different areas of the asset.

The theme's findings from the semi-structured interviews concur with those of the literature review (Section 2). The interview participants viewed digital twins as a two-way interaction between a virtual model and physical asset, which has been highlighted by several authors [12,30]. This is also in agreement with Madni et al. [26], who further elaborated on the nature of the data link and defined various modes of digital twins. The finding of the IoT-enabled devices and sensor-based data-capturing devices as being the key enablers for digital twins conforms also with the literature. where the emergence of digital twins is attributed to data-driven digital technologies, such as cloud computing, IoT, AI, and big data analytics [35,43,44,59]. Hence, it can be concluded that while an agreement about the definition of a digital twin is lacking, there is a consensus that data-centric technologies provide opportunities to extend the capabilities of the current 3D model (or BIM) to capture the user behaviors, relationships, and data link between spatial entities in the physical world and a virtual model, thus underpinning the development of digital twins for built-environment projects.

### 4.2. Theme 2: Applications and Benefits of Digital Twins

The applications of digital twin in the built environment mentioned by the participants are across construction projects, manufacturing operations, smart cities, healthcare projects, and the mass procurement and maintenance of assets and estates or asset portfolios.

There was a clear split among the participants about digital twin applications within the design and construction stage. One group (three participants) opined that a digital twin could improve the design and construction process. It was quoted that "BIM is a tool for design; however, this is definitely not true for digital twins." Further, they considered that digital twins could facilitate design improvements owing to their data-driven ability for presenting "what-if" analyses that may be applied to studies related to the lighting, heating, space management, and functional workflow of building projects. It was also argued by two participants that a digital twin cannot be created in the pre-construction design phase without a physical asset to replicate. However, they considered such applications of a digital twin as design "optioneering" tools or prototypes that could have the capability to test design options against contextual or anticipated data. Three participants suggested potential applications of digital twins in the design and construction phases, including the creation of a virtual model and incorporating real-time sensor monitoring to create alerts for critical site management issues such as safety management (e.g., notifying supervisors about the need to undertake safety measure if workers wander into restricted or danger-

ous zones) where digital twins may play a role in reducing the number of accidents at construction sites. Digital twins, according to three participants, can improve the quality of design, which would imply a reduced number of requests for information queries and design changes and less rework on construction sites.

This group of participants (No. 3) supported the claims made by Çıdık et al. [45] and Rasheed et al. [41] who argued that modeling BIM technologies cannot accommodate the dynamism required for design development and improvement. Several other authors have supported digital twin applications for design improvement through simulations and compliance checking [8,25,46,47]. However, this view was contradicted by two participants, who argued that although a digital twin can provide and maintain a comprehensive 3D model, it cannot improve the design and construction processes. In their opinion, digital twins are mainly employed in the asset management stage and not at the design or construction stage. Further, it was argued that an amalgamation of various discipline models (architectural, structural, MEP, specialty equipment, etc.) and the linking of a live data feed from physical to virtual assets during the construction stage is not yet practically achievable with the existing tools and processes.

The participants unanimously agreed on the potential applications of digital twins for asset management within the operation phase of an asset's lifecycle. The participants stated that digital twins, using component data such as the type, manufacturer, supplier, warranty, and maintenance schedules together with their actual operational performance, would help predict and manage failures and shape the maintenance policy for future operations. Participants stated that a "digital twin would create alerts for the due maintenance activities or parts replacements" and "it would avoid potential mechanical failures at airports or shopping malls". Three of the participants further elaborated by outlining how a digital twin could correlate data sets such as asset usage or environmental impact to simulate and predict necessary maintenance or to implement mitigation measures. However, two participants expressed concerns about linking such information to the existing building management systems, stating that present building automation systems are not linked with FM data and systems, and hence they cannot augment the operational activities. These results concur with the literature findings on the applications of digital twins for proactive asset management [7,38].

### 4.3. Theme 3: Implementation Challenges

Eight participants agreed that a cultural change in the built environment sector is essential for fostering digitalization, including the adoption of digital twins. The participants unanimously acknowledged that there is a need to change the working practices of the construction industry to embrace new and emerging technologies and processes. However, cultural change is slow due to a number of reasons highlighted in the interviews, such as limited investment in innovation, the reluctance of professionals, inadequate senior level buy-in, lack of sufficient proven benefits of digitalization, an unwillingness from organizations to assume the risk associated with implementing innovation, and the transient nature of supply chains and project teams. These resonate with the recommendations of several authors stressing the need to upskill the construction sector workforce with technical competencies (e.g., use of digital tools) and non-technical competencies to transform the overall industry [12,41,63,64].

All ten participants agreed that a key challenge in adopting digital twins in the built environment sector is the lack of "an all-inclusive toolset" as highlighted by Adams [65] and Cureton [40], who argue that cost-effectiveness through successful case studies is essential for the uptake of new technologies. Two participants further elaborated on the development of case studies to demonstrate the benefits of digital twin. Stating that while they would be highly beneficial in evidencing the value of digital twins to an organization's senior management, the creation of a comprehensive set of case studies could take years to compile. This was thought to be due to the myriad of potential applications and purposes that digital twins could be applied to.

It was highlighted by three participants that the existing digital tools require significant customization, have interoperability issues, and offer solutions in combination with several tools and software applications, which add to the complexity and cost and result in a steep learning curve for widespread adoption in the industry. There was also a sense of reticence put forward by three of the interviewees concerning the issue of organizations being lured into long-term software subscriptions which would remove some of their agility in innovation.

The industry lacks compatible and adequate data standards [21,65] not only between the stages of an asset's lifecycle but also within the same stage and even within the same project enterprise. This is a major challenge for the applications of digital twins identified earlier, as many of these applications (e.g., facilities management, smart cities operation) need data to be collected and shared from different project stages, project actors (e.g., designers, manufacturers, constructors) and tools. This challenge will hinder information sharing to the extent that the participants (No/. 2) opined that it may take another decade to streamline the required technical solutions to digitalize the built-environment industry effectively through digital twins. This is also in agreement with the literature findings, where interoperability was highlighted by Monsone et al. [58] and Qi et al. [48] as a key challenge for the future development of digital twins. With the absence of common standards, digital twin application for FM operations would remain ineffective according to Re Cecconi et al. [37] who also agree that this is a fundamental aspect for the successful implementation of digital twins in the built environment.

Six participants expressed concerns about data security, which they considered a crucial challenge to the adoption of digital twins in construction projects. One participant stressed that with existing IoT devices data breach is more likely than without such devices. This observation was further validated by three other participants, who agreed that despite the advances in information and computing technologies, complete data security cannot be guaranteed. In addition to the security challenge, the issue of ownership and the sharing of data arose with three participants who suggested this issue will be particularly significant on the client side, as clients would require an understanding as to what data and how data can be shared in a digital twin. These issues were highlighted by Brozohvsky et al. [54] and Monsone et al. [58], who reported that privacy and the ownership of data are the primary issues with regard to the adoption of digital twins in the industry.

Despite the challenges identified, seven of the participants were in agreement that with the current pace of technological advancements, the organizational and industry aspirations for innovation, and existing technical capabilities of the construction industry, the development and implementation of digital twins in the industry is becoming ever more possible.

### 4.4. Theme 4: Existing Practical Experiences of Digital Twins

Seven of the participants agreed that although digital twin technology has anticipated advantages in built-environment applications such as in asset management, their practical applications are still rare. Two participants were contrary to this position, giving examples of technology deployment and data utilization that they felt adhered to their definitions of a digital twin, both of which were in the water utility sector and were for similar purposes (water flow management). Four of the other participants regarded the current asset management practices as mainly document-oriented and reactive, which can be transformed into proactive maintenance tasks using digital twins, resulting in more efficient and cheaper FM operations. Three participants also predict that despite their potential, digital twins will be embraced by large clients and government institutions before they can be adopted by the majority of clients and operators within the built environment. They stressed the need for examples of practical implementation with the return-on-investment figures as well as robust business cases that demonstrate value in an already recognized format to aid in convincing clients to invest in digital twin adoption and reap potential benefits. They further stated that obtaining the right data for integration with the CAFM

system is regarded as a huge challenge and a slow process, and its use has been limited to specific areas of asset management and needs to be experienced on a larger scale.

These findings are a clear indication that due to the relative newness of the concept of digital twins in the built environment sector, their practical applications have been limited to prototypes and experimental and research implementation thus far. These findings also resonate with those of Ruohomaki [44] who claims that digital twin implementation on a wider scale has not been observed owing to a lack of technological support, including the inefficient integration and communication of systems.

### 4.5. Theme 5: Future Improvement Suggestions and Timeline

The participants unanimously agreed that digital twins would eventually be widely applied in the built-environment sector, but as in the case of any new digital intervention, their usefulness must be proved before a majority of the stakeholders in the industry can support their adoption. The participants stressed that "seeing is believing", stating that the industry needs real-world examples of digital twins with proven benefits and return on investment to give it serious consideration. At present, only theoretical knowledge of digital twins in the built environment is available, and information/research on their practical implementation is scarce. Three participants also mentioned the potential to learn and demonstrate benefits with examples from industries outside the built environment, such as manufacturing and aerospace, where the application of digital twins is more mature.

Six participants also expressed the need for an awareness campaign through the academic literature, conferences, and industry events to educate the industry about the potential and applications of digital twins. They also noted the importance of social media and online forums in the dissemination and challenges of digital twin research, as it was thought that this played a major role in the development of BIM. Thought to be an essential part of the development of digital twins, four of the participants advocated that there is a need to define their relationship with existing forms of 3D model representations of data.

Two participants perceived the implementation process of digital twin systems as complex, which would undermine the benefits associated with digital twins and its business case cost-effectiveness, as employers would be reluctant to invest in digital twin applications. Digital twins in the construction sector are also facing the challenge of attracting sufficient funding for research and development, an issue that is exacerbated by the limited availability of successful case studies to substantiate the benefits to be gained from their application [40,66]. The current rate of adoption is greatly attributed to the cultural changes required at a personal as well as at an organizational level [8,67]. There is a large portion of professionals within the built environment field that find their existing conventional structure for executing construction projects more practical owing to familiarity; hence, the adoption of a new structure would require some convincing [13]. In addition to the commercial and technical obstacles that an organization must overcome to upgrade their systems, the lack of incentives makes it more difficult for organizations to consider the adoption of digital twins [8]. In relation to this challenge, seven participants indicated that, as in the case of BIM, the development and implementation of digital twins would benefit from the support of large client bodies, governments, and legislative institutions to provide large path finder projects and studies, as well as overarching principles for the creation of digital twins. At an organization level, three participants claimed there is a need for buy-in at multiple levels, including from senior management and the "boots on the ground" level, so digital twins may become part of an organization's objectives and working practices.

## 5. Conclusions

A digital twin is a nascent concept within the built environment. A consensus about the definition of digital twins is still lacking but there is an agreement that data-centric technologies under the digital twin concept can provide opportunities to extend the capabilities of the current BIM to capture behaviors and relationships and develop a new breed of data-centric decision-making process. Although proven business cases and working

examples of digital twins are limited, digital twins are considered by the literature and interviewed experts to have the potential to bring both new applications and benefits to the built environment sector.

The applications of digital twins can benefit different processes within built-environment projects across the design, construction, and operation. However, the implementation of digital twins in the built environment is still at an early research and development stage in both industry and academia, and demonstration in real-world industrial projects at scale has yet to be seen.

The challenges facing the implementation of digital twins are related to the availability of technology and the complexity of technological systems constituting digital twins; lack of common data standards and tools, data security, and ownership; workforce upskilling; and the necessity for systemic cultural change. The similarities that can be drawn between digital twins and BIM in their development and adoption mean that digital twins can learn lessons from the challenges that were faced in the development of BIM. Learning from different industry sectors such as aerospace and manufacturing where the adoption of digital twins is more diffused can benefit their implementation in the built environment. Finally, providing empirical and rigorous evidence about the benefits and advantages of digital twins from real use cases was identified as necessary for their future adoption.

**Author Contributions:** Conceptualization, M.S., M.K. and M.T.S.; Methodology, M.S., M.T.S., M.K. and D.D.; Writing—original draft, M.S., M.K. and M.T.S.; Investigation, M.S. and D.D., Formal analysis, M.S., D.D.; Supervision M.K. and M.T.S.; Writing—review & editing, M.K. and M.T.S.; Project administration, M.K. and M.T.S. All authors have read and agreed to the published version of the manuscript.

**Funding:** The APC was funded by the UAE University, under the grant 31N397-UPAR (5) 2019.

**Data Availability Statement:** Not Applicable.

**Acknowledgments:** Authors are thankful to all interview participants for supporting this research.

**Conflicts of Interest:** The authors declare no conflict of interest.

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
