# Peer review of "Digital Twins in Built Environments: An Investigation of the Characteristics, Applications, and Challenges"

_buildings, doi:10.3390/buildings12020120_

Round 1

Reviewer 1 Report

In the introduction, should the authors deliver a more recent and deep discussion on the stage/levels of a Digital Twin, search by keywords as; incremental twin, digital shadow, DT evolution, platforms generation. 
Also, in my perspective, it is not appropriate for an academic paper to have reproduced Figures from other Authors. I would suggest coming up with the authors' own Figures based on the references; in this, get into account to review Figures 1 and 2.
At the literature review, as it is not a scoping review as it should be, it is pretty impossible do not to point some gaps (e.g., Boje, C., Guerriero, A., Kubicki, S., & Rezgui, Y. (2020). Towards a semantic Construction Digital Twin: Directions for future research. Automation in Construction, 114(November 2019), 103179. https://doi.org/10.1016/j.autcon.2020.103179). If the authors want to improve the paper quality, I advise conducting a scoping review.
Also, the Authors should include discussions in Table 2 about the DTC (Digital Twin Construction) by Sacks, R., Brilakis, I., Pikas, E., Xie, H. S., & Girolami, M. (2020). Construction with digital twin information systems. Data-Centric 837 Engineering, 1. As it is also a new vision from Professor Brilakis, as well co-author of that paper.
Many long sentences overall manuscript should be rewritten to increase readability. Also, the reference style seems imprecise.
Finally, the conclusions should highlight the Characteristics, Applications and Challenges in separate paragraphs or use some bullet points.

Author Response

We are thankful to the reviewer for providing valuable insight into our work. We agree with the reviewer. Figure 1 is removed, as it was not adding much value in the continuity of the discussion. We have reproduced and extended figure 2 in line with the text discussion. Thank you for highlighting this. We have rechecked and revised the overall manuscript. We hope that the submitted revision has addressed all the reviewer’s concerns and has incorporated valuable suggestions.

Thank you once again.

Reviewer 2 Report

The concept of the digital twin is proposed as a new technology-led advancement to support the processes of design, construction, and operation of built assets. Although a consensus about its definition is still lacking, most scholars perceive it as a digital or cyber environment that is bidirectionally-linked to its physical or real-life replica to enable simulation and data-centric decision making. Studies have started to investigate its role in the digitalization of asset delivery including the management of built assets at different levels within the building and infrastructure sectors. However, questions persist regarding its actual applications and implementation challenges including its integration with other digital technologies (i.e., Building Information Modeling, Virtual and Augmented Reality, Internet of Things, Artificial Intelligence, and Cloud Computing). This paper can be accepted with minor revision. please provide some quantitative indexes in the abstract. For figure 1, the authors cited Smartindustry, 2017, I think it is older. Future works had better be added in conclusion.

Author Response

Thank you for your feedback.

The highlighted reference” Smartindustry, 2017” has been removed.

We have rechecked and revised the overall manuscript. We hope that the submitted revision has addressed all the reviewer’s concerns and has incorporated valuable suggestions.

Thank you once again.

Reviewer 3 Report

It is an interesting paper that addresses a relatively new topic, which conceptually is expected to have a very positive impact on the AEC industry. The paper is very well written, with abundant and varied references, which account for an adequate review of the background of the subject.
The methodology is clear and concise and is well explained throughout the paper.
The paper makes an in-depth study of the characteristics of digital twins, however, I consider that the main characteristics should be summarized in the conclusions, as has been done with the challenges and with the applications.

Author Response

Dear Reviewer

Thank you for your feedback – much appreciated.

We have made several changes in the revised submission, following suggestions from the reviewers, which have improved the overall manuscript.

Thank you.

Reviewer 4 Report

The article provides a comprehensive literature review of digital twin definitions in connection with modeling of built environment, the comparison of the BIM concept and the digital twin concept with the discussion of the differences and overlaps of these concepts, the review of expectations of digital twin use possibilities in various AECO fields, and the results of the research among 10 specialists from the AECO industry and compares these results with the findings in the literature.

The article provides highly valuable results, but I can recommend the inclusion of the following comments.

1. The article deals with the term ‘built environment’, but does not provide a definition of the term. The term ‘built environment’ has various definitions similarly as the term ‘digital twin’. The authors should provide the used definition of the term.

2. From the provided ‘digital twin’ definition can be seen that there is still no consensus on whether the term ‘digital twin’ should be used in the sense of data (model of the built environment) or should also include the processes of collecting, storing, transmitting, processing and interpreting these data. Provided ‘digital twins’ definitions can be distinguished based on this point of view.

3. The authors compare the term ‘digital twin’ with the term ‘BIM’. The abbreviation ‘BIM’ can mean ‘Building information model’ (data) or ‘Building information modeling’ (process). I am omitting the less common uses of the BIM abbreviation as ‘Building information management’, ‘Better information management’, etc. Therefore, the authors should clearly state where the BIM abbreviation is used as ‘Building Information Modeling’ (e.g, l. 34) and ‘Building Information Model’ (e.g. l. 77-79).

4. Most of the digital twin definitions and BIM definitions presented use fundamental terms of the systems engineering domain (mainly the terms ‘system’ and ‘model’). Definitions of these terms should be provided. The compliance of the provided ‘digital twin’ definitions with these definitions can be discussed. The following sources can be used:

[1] ISO/IEC/IEEE 24765:2010(E) - Systems and software engineering — Vocabulary

[2] INCOSE, and Wiley. INCOSE Systems Engineering Handbook : A Guide for System Life Cycle Processes and Activities, John Wiley & Sons, Incorporated, 2015. ProQuest Ebook Central, https://ebookcentral.proquest.com/lib/cvut/detail.action?docID=4040424.

[3] Buede, Dennis M., and William D. Miller. The Engineering Design of Systems : Models and Methods, John Wiley & Sons, Incorporated, 2016. ProQuest Ebook Central, https://ebookcentral.proquest.com/lib/cvut/detail.action?docID=4391537.

Author Response

Thank you for your feedback – much appreciated.

Thank you for highlighting these gaps. We have added a definition for “built environment’ and have replaced “building information models” and “building information modelling” where it may have caused confusion for the readers.

We have rechecked and revised the overall manuscript. We hope that the submitted revision has addressed all the reviewer’s concerns and has incorporated valuable suggestions.

Thank you once again.

Round 2

Reviewer 1 Report

The authors did not address issues pointed in the first review. As well, there is not provide a structured response to the review made. Based on that, I would like to point it again, for at least get a response from the Authors.

1. In the introduction, should the authors deliver a more recent and deep discussion on the stage/levels of a Digital Twin, search by keywords as; incremental twin, digital shadow, DT evolution, platforms generation. 

2. Also, in my perspective, it is not appropriate for an academic paper to have reproduced Figures from other Authors. I would suggest coming up with the authors' own Figures based on the references; in this, get into account to review Figures 1 and 2. It was the only response I had.

3. At the literature review, as it is not a scoping review as it should be, it is pretty impossible do not to point some gaps (e.g., Boje, C., Guerriero, A., Kubicki, S., & Rezgui, Y. (2020). Towards a semantic Construction Digital Twin: Directions for future research. Automation in Construction, 114(November 2019), 103179. https://doi.org/10.1016/j.autcon.2020.103179). If the authors want to improve the paper quality, I advise conducting a scoping review.

4. Also, the Authors should include discussions in Table 2 about the DTC (Digital Twin Construction) by Sacks, R., Brilakis, I., Pikas, E., Xie, H. S., & Girolami, M. (2020). Construction with digital twin information systems. Data-Centric 837 Engineering, 1. As it is also a new vision from Professor Brilakis, as well co-author of that paper.

5. Many long sentences overall manuscript should be rewritten to increase readability. Also, the reference style seems imprecise.

6. Finally, the conclusions should highlight the Characteristics, Applications and Challenges in separate paragraphs or use some bullet points.

Author Response

Reviewer Comments

Authors response

The authors did not address issues pointed in the first review. As well, there is not provide a structured response to the review made. Based on that, I would like to point it again, for at least get a response from the Authors.

1. In the introduction, should the authors deliver a more recent and deep discussion on the stage/levels of a Digital Twin, search by keywords as; incremental twin, digital shadow, DT evolution, platforms generation. 

2. Also, in my perspective, it is not appropriate for an academic paper to have reproduced Figures from other Authors. I would suggest coming up with the authors' own Figures based on the references; in this, get into account to review Figures 1 and 2. It was the only response I had.

3. At the literature review, as it is not a scoping review as it should be, it is pretty impossible do not to point some gaps (e.g., Boje, C., Guerriero, A., Kubicki, S., & Rezgui, Y. (2020). Towards a semantic Construction Digital Twin: Directions for future research. Automation in Construction, 114(November 2019), 103179. https://doi.org/10.1016/j.autcon.2020.103179). If the authors want to improve the paper quality, I advise conducting a scoping review.

4. Also, the Authors should include discussions in Table 2 about the DTC (Digital Twin Construction) by Sacks, R., Brilakis, I., Pikas, E., Xie, H. S., & Girolami, M. (2020). Construction with digital twin information systems. Data-Centric 837 Engineering, 1. As it is also a new vision from Professor Brilakis, as well co-author of that paper.

5. Many long sentences overall manuscript should be rewritten to increase readability. Also, the reference style seems imprecise.

6. Finally, the conclusions should highlight the Characteristics, Applications and Challenges in separate paragraphs or use some bullet points.

1.       We feel it is too early to discuss these concepts in the introduction. However, these are covered in the literature review. See lines 202-213. The purpose of our paper is not to discuss in depth the level but to focus on applications, challenges and enabling technologies as stated in the abstract. We acknowledge that enabling challenges would vary across different level of DT but this specialised discussion is beyond the scope of this paper.

2.       The image was adapted. Nevertheless, we have not created our own image.

3.       We have cited this works and we are aware of the systematic and scoping review. As this paper contains mixed methods (literature review, 10 interviews), adding a scoping or a systematic review would be a significant step forward that is beyond the scope of the paper. A systematic or scoping review can be published by its own right without the interviews. However, we are confident that most relevant works have been captured and used in this paper.

4.       DTC work by Sacks and Brilakis is well known and has been cited both in the table and the body of the paper.

5.       The paper was reviewed for readability and grammar.

6.       The conclusions have been structure as suggested – Thank you for your valuable feedback.

Round 3

Reviewer 1 Report

The Authors said it "4. DTC work by Sacks and Brilakis is well known and has been cited both in the table and the body of the paper." Where in the Tables Sacks DTC works was presented? 
It is so bad that false statements; I hope all Authors are aware of these actions.